# Finding Mixed Strategy Nash Equilibrium for Continuous Games through Deep Learning

## Abstract

Nash equilibrium has long been a desired solution concept in multi-player games, especially for those on continuous strategy spaces, which have attracted a rapidly growing amount of interests due to advances in research applications such as the generative adversarial networks. Despite the fact that several deep learning based approaches are designed to obtain pure strategy Nash equilibrium, it is rather luxurious to assume the existence of such an equilibrium. In this paper, we present a new method to approximate mixed strategy Nash equilibria in multi-player continuous games, which always exist and include the pure ones as a special case. We remedy the pure strategy weakness by adopting the pushforward measure technique to represent a mixed strategy in continuous spaces. That allows us to generalize the Gradient-based Nikaido-Isoda (GNI) function to measure the distance between the players' joint strategy profile and a Nash equilibrium. Applying the gradient descent algorithm, our approach is shown to converge to a stationary Nash equilibrium under the convexity assumption on payoff functions, the same popular setting as in previous studies. In numerical experiments, our method consistently and significantly outperforms recent works on approximating Nash equilibrium for quadratic games, general blotto games, and GAMUT games.

## 1 Introduction

Nash equilibrium (Nash, 1950) is one of the most important solution concepts in game scenario with multiple rational participants. It plays an important role in theoretical analysis of games to guide rational decision-making processes in multi-agent systems. With the recent success of machine learning applications in games, it attracts even more research interests on applying machine learning technique for unsolved game theory problems, for example, computation of Nash equilibrium for multi-player games. In this paper, we focus on games with continuous action spaces, which include the famous application for Generative Adversarial Networks (GANs) (Goodfellow et al., 2014), as well as many important game types such as the colonel blotto game (Gross & Wagner, 1950), Cournot competition (R, 1996). We develop a solution which significantly improves the status-quo.

There have been several successful approaches to compute Nash equilibrium for multi-player (mostly 2-player) continuous game (Raghunathan et al., 2019; Balduzzi et al., 2018; Letcher et al., 2018). These works seek Nash equilibria corresponding to pure strategies, in which each player takes a specific action to achieve its best payoff given other players' actions. A major concern for such a solution concept is its possible non-existence. As a result, the convergences to a Nash equilibrium for these approaches were proven under the assumption for the existence of a pure strategy Nash equilibrium, which can hardly be checked in practice, and their applicability is limited to specific types of games. On the contrary, it is known that mixed strategy Nash equilibria always exist under mild conditions. And note that any pure strategy Nash equilibrium is also a mixed strategy Nash equilibrium, which means the latter one is a much more desired solution concept.

However, a key challenge that obstructs the study of computing a mixed strategy Nash equilibrium, especially for a continuous game, lies on how to design an efficient method to represent the mixed strategy. To be precise, a pure strategy can be represented by a single variable choosing from some region. But as a distribution on each player's action space, a mixed strategy with respect to the player is defined in a (subspace of) real space $\mathbb{R}$. More generally, exact representation for a mixed strategy of a player usually requires many variables in a continuous space. In addition, the corresponding probability distribution may not have a density function in closed-form.

To address this challenge, we introduce a pushforward measure technique. It is a common tool in measure theory to transfer a measure to some specific measure space (Bogachev, 2007). Specific to a continuous game, the probability distribution corresponding to a mixed strategy is obtained via a mapping parameterized by neural nets from a multi-dimensional uniform distribution.

With this pushforward representation, we generalize the Gradient-based Nikaido-Isoda (GNI) function introduced by Raghunathan et al. (2019), to handle mixed strategy Nash equilibria. The original GNI function can be viewed as a measure for the distance between any joint strategy profile and a Nash equilibrium after applying the payoff functions of players. With proper generalization and modification, we develop its mixed strategy version as a proper measure for a Nash equilibrium. We prove that the distance becomes zero if and only if a stationary mixed Nash equilibrium is obtained. Then we apply the gradient descent algorithm to the general GNI function, which converges to a stationary mixed Nash equilibrium under the convexity assumptions on the payoff functions.

Finally, we compare our method with baseline algorithms in numerical experiments. Our approach shows effective convergence property in all the randomly generated quadratic games, general blotto games, GAMUT games and Delta-Dirac GAN (in appendix), which outperforms other baselines.

## 2 BACKGROUND AND PROBLEM DESCRIPTION

The discrete action space Nash equilibrium computation has been most widely studied in the literatures. Most well-known being the Lemke–Howson algorithm (Lemke & Howson, 1964) for solving the bimatrix game. The state-of-art theoretical work of Tsaknakis & Spirakis (2007) provided a solution of $1/3$ approximation in polynomial time . And empirical work (Fearnley et al., 2015) shows it also performs well against practical game solving methods for the bimatrix game.

However, the continuous action space game computation is widely used in practice, while few methods are known for the general Nash equilibrium computation and all restricted to pure strategies. The GNI function is considered by Raghunathan et al. (2019) as one possible technique for multi-player continuous game, but they only consider pure strategies. With further assumption that the utility functions of players are twice continuous differentiable, gradient-based algorithms are developed (Balduzzi et al., 2018; Letcher et al., 2018). But they do not provide a way to parameterize mixed strategy of continuous action space, and even we are able to do so through neural networks, the Hessian of utility functions, necessary in these approaches, is extremely hard to compute or store.

Game-theoretical approach has had useful applications to machine learning such as the optimization of GAN network training (Daskalakis et al., 2017; Gidel et al., 2018) and adjustment on the gradient descent method (Balduzzi et al., 2018). However they are limited to 2-player zero-sum games.

We are the first work to study the mixed strategy continuous game Nash equilibrium computation. Our work is motivated by the utilization of the Nikaido-Isoda (NI) function for loss function minimization (Uryas' ev & Rubinstein, 1994; Raghunathan et al., 2019). We start to establish a theoretical formulation of the extend mixed strategy continuous action space Nash equilibrium as a result of the minimization on a functional variation-based Nikaido-Isoda function.

### 2.1 CONTINUOUS GAME NASH EQUILIBRIUM

$$\text{Find } \mathbf{x}^* = (x_1^*, x_2^*, \cdots, x_N^*)$$
$$\text{s.t. } x_i^* = \arg \min_{\mathbf{x} \in \mathbb{R}^n : \mathbf{x}_{-i} = \mathbf{x}_{-i}^*} f_i(\mathbf{x}) \tag{1}$$

Here $N$ denotes the number of players, and $x_i \in \mathbb{R}^{n_i}$ the strategy of the $i$-th player where $n_i$ is the dimension of its action space. Let $n = \sum_{i=1}^N n_i$, and $\mathbf{x} = (x_1, x_2, \cdots, x_N) \in \mathbb{R}^n$ denotes the joint pure strategy among all players while $\mathbf{x}_{-i} = (x_1, \cdots, x_{i-1}, x_{i+1}, \cdots, x_N) \in \mathbb{R}^{n-n_i}$ the joint pure strategy among players except $i$. $f_i : \mathbb{R}^n \to \mathbb{R}$ denotes the utility function (cost) of $i$-th player. A solution $\mathbf{x}^*$ to (1) is called a pure strategy Nash equilibrium.

### 2.2 NIKAIDO-ISODA (NI) FUNCTION

In the paper by Nikaidô et al. (1955), Nikaido-Isoda (NI) function is introduced as:

$$\phi(\mathbf{x}) = \sum_{i=1}^N \left( f_i(\mathbf{x}) - \inf_{\hat{\mathbf{x}} \in \mathbb{R}^n : \hat{\mathbf{x}}_{-i} = \mathbf{x}_{-i}} f_i(\hat{\mathbf{x}}) \right) \triangleq \sum_{i=1}^N \phi_i(\mathbf{x}) \tag{2}$$

From the Equation (2), we know $\phi(\mathbf{x}) \geqslant 0$ for $\forall \mathbf{x} \in \mathbb{R}^n$, and $\phi(\mathbf{x}) = 0$ is the global minimum of NI function which can only be achieved at a Nash equilibrium (NE). Therefore, a common algorithm of computing NE points is minimizing the NI function above. However, it is a huge difficulty to handle the global infimum. On one hand, the global infimum can not be obtained in finite time. On the other hand, the infimum can be unbounded below in some games, for example the two-player bi-linear games, where $f_1(\mathbf{x}) = x_1^T M x_2 = -f_2(\mathbf{x})$. All of the facts above show us the shortcomings of NI function, and in order to rectify them, Raghunathan et al. (2019) introduce the following Gradient-based Nikaido-Isoda (GNI) function.

### 2.3 GRADIENT-BASED NIKAIDO-ISODA (GNI) FUNCTION

If we calculate local infimum in the NI function $\phi(\mathbf{x})$ instead of global infimum, the time complexity and unbounded infimum are no longer shortcomings. To be precise, given the local radius $\lambda$, local infimum can be approximated by steepest descent direction, and we get the following GNI function:

$$V(\mathbf{x}; \lambda) = \sum_{i=1}^{N} \left( f_i(\mathbf{x}) - f_i(x_1, \cdots, x_{i-1}, x_i - \lambda \nabla_i f_i(\mathbf{x}), x_{i+1}, \cdots, x_N) \right)$$

By minimizing $V(\mathbf{x}, \lambda)$, a stationary Nash point $\mathbf{x}^*$, where $\nabla_{x_i} f_i(\mathbf{x}^*) = 0$ for $\forall i$, can be approximated efficiently. Furthermore, if all the utility functions $f_i$ are convex, then the stationary Nash points (SNP) obtained are actually Nash Equilibrium (NE).

## 3 (MIXED-GNI) GRADIENT-BASED NIKAIDO-ISODA FUNCTION OF MIXED STRATEGY ON CONTINUOUS GAMES

In this section, we are going to introduce our novel Gradient-based Nikaido-Isoda function of mixed strategy on continuous games (Mixed-GNI), which is used to get an approximated solution of the following optimization problem.

$$\begin{aligned}
&\text{Find } \pi^* = (\pi_1^*, \pi_2^*, \cdots, \pi_N^*) \\
&\text{s.t. } \pi_i^* = \arg\min_{\pi:\pi_{-i}=\pi_{-i}^*} \mathop{\mathbb{E}}_{x_j \sim \pi_j, \, \forall j} f_i(x_1, x_2, \cdots, x_N)
\end{aligned} \tag{3}$$

Before we solve this optimization problem, there is another fundamental question, which is how we should represent (or parametrize) a distribution $\pi_i$. The simplest way to do so is to parametrize its density function. However, not every distribution has its density function, such as Dirac distribution, and it will be inconvenient for us to do sampling from only a density function. Therefore, we introduce another way, adopting the pushforward measure to represent a distribution.

Given a distribution $\mu_0$ and a mapping $g(\cdot)$, data $\mathbf{x}$ drawn from $\mu_0$ can be transported into a new distribution $\mu_1$ (constituted by $g(\mathbf{x})$). Technically speaking, $\mu_1$ is called the pushforward measure of $\mu_0$ by mapping $g$, denoted by $\mu_1 = g^\#(\mu_0)$.

Here, for $\forall j \in [N]$, we prepare each distribution $\pi_j$ a corresponding pushforward function $g_j : \mathbb{R}^d \to \mathbb{R}^{n_j}$, and we have:

$$\pi_j = g_j^\#(U)$$

where $U$ stands for the uniform distribution on $[0,1]^d$. Each time we want to sample from distribution $\pi_i$, we only need to sample several $\omega_i \in [0,1]^d$ from distribution $U$ and calculate $g_i(\omega_i)$. Then, these $g_i(\omega_i)$ form a sample set from distribution $\pi_i$. And optimization problem (3) becomes:

$$\begin{aligned}
&\text{Find } \mathbf{g}^* = (g_1^*, g_2^*, \cdots, g_N^*) \\
&\text{s.t. } g_i^* = \arg\min_{\mathbf{g}:g_{-i}=g_{-i}^*} \mathop{\mathbb{E}}_{\omega_j \sim U, \, \forall j} f_i(g_1(\omega_1), g_2(\omega_2), \cdots, g_N(\omega_N))
\end{aligned} \tag{4}$$

To solve the optimization problem above, we consider the following Gradient-based Nikaido-Isoda function of Mixed strategy on Continuous games (Mixed-GNI), generalized from the GNI function introduced above, and we call this function $V$ the **local regret**:

$$\begin{aligned}
V(g_1, g_2, \cdots, g_N; \lambda) &= \sum_{i=1}^{N} F_i(g_1, g_2, \cdots, g_N) - F_i(g_1, \cdots, g_{i-1}, g_i - \lambda \delta_{g_i} F_i, \cdots, g_N) \\
&\triangleq \sum_{i=1}^{N} V_i(g_1, g_2, \cdots, g_N; \lambda)
\end{aligned} \tag{5}$$

Here, $\delta_{g_i} F_i$ stands for the 1-st order variation of functional $F_i$ on element function $g_i$ and

$$F_i(g_1, g_2, \cdots, g_N) = \mathbb{E}_{\omega_j \sim U, \ \forall j} [f_i(g_1(\omega_1), g_2(\omega_2), \cdots, g_N(\omega_N))]$$

By minimizing the functional $V(g_1, g_2, \cdots, g_N; \lambda)$, we can approximately get stationary Nash points (SNP), and even get Nash equilibrium if all the utility functions $f_i$ are convex. We will prove them in the next section.

In practice, we further parametrize these pushforward functions as: $g_i(\cdot) = g_i(\cdot, \theta_i)$, to efficiently calculate derivatives instead of variations. For simplicity, we denote $g_i$ as $g_{\theta_i}$. In order to obtain a better expressibility, we use neural networks as the architecture to parametrize these pushforward functions. Then, Mixed-GNI function $V$ can be transformed to:

$$V(g_{\theta_1}, g_{\theta_2}, \cdots, g_{\theta_N}; \lambda) = \sum_{i=1}^{N} F_i(g_{\theta_1}, g_{\theta_2}, \cdots, g_{\theta_N}) - F_i(g_{\theta_1}, \cdots, g_{\theta_{i-1}}, g_{\theta_i - \lambda \partial_{\theta_i} F_i}, \cdots, g_{\theta_N})$$

Finally, the Mixed-GNI function can be estimated by sampling the points from distribution $U$ as an estimator of these $F_i$ and minimized by implying gradient descent on these function parameters $\theta_i$, $i \in [N]$, the convergence of which is proved in the next section.

# 4 THEORETICAL ANALYSIS OF MIXED-GNI

## 4.1 THE SUFFICIENT AND NECESSARY CONDITION OF STATIONARY NASH POINT

As a mixed strategy of an $N$-player continuous game, $(\pi_1, \pi_2, \cdots, \pi_N) = (g_1^\# U, g_2^\# U, \cdots, g_N^\# U)$ is a stationary Nash point (SNP) if and only if for $\forall i \in [N]$, the 1-st order variation

$$\delta_{g_i}(F_i)[\sigma(x)] = 0 \tag{6}$$

holds at each direction $\sigma(x)$. Here:

$$F_i(g_1, g_2, \cdots, g_N) = \mathbb{E}_{\omega_j \sim U, \ \forall j}[f_i(g_1(\omega_1), g_2(\omega_2), \cdots, g_N(\omega_N))]$$

is the expectation of the $i$-th player's utility with the form of $N$-variable functional. Now, we compute the variation above and deduce the sufficient and necessary condition of SNP.

$$\begin{aligned}
\delta_{g_i}(F_i)[\sigma(x)] &= \lim_{\epsilon \to 0} \frac{1}{\epsilon} \left( F_i(g_1, g_2, \cdots, g_N) - F_i(g_1, \cdots, g_i - \epsilon\sigma, \cdots, g_N) \right) \\
&= \mathbb{E}_{\omega_j \sim U, \ \forall j}[\sigma(\omega_i)^T \cdot \nabla_i f_i(g_1(\omega_1), g_2(\omega_2), \cdots, g_N(\omega_N))] \\
&= \mathbb{E}_{\omega_i \sim U}[\sigma(\omega_i)^T \cdot \mathbb{E}_{\omega_j \sim U, \ \forall j \neq i}[\nabla_i f_i(g_1(\omega_1), g_2(\omega_2), \cdots, g_N(\omega_N))]] \\
&\triangleq \mathbb{E}_{\omega_i \sim U}[\sigma(\omega_i) \cdot G(\omega_i)] = \int_{[0,1]^d} \sigma(\omega_i) \cdot G(\omega_i) d\omega_i
\end{aligned} \tag{7}$$

where:

$$G(\omega_i) = \mathbb{E}_{\omega_j \sim U, \ \forall j \neq i}[\nabla_i f_i(g_1(\omega_1), g_2(\omega_2), \cdots, g_N(\omega_N))]$$

For SNP, Equation (6) holds at each direction $\sigma(x)$, i.e. $G(\omega_i) \equiv 0$. Therefore, we have

**Theorem 1.** $\pi = (\pi_1, \pi_2, \cdots, \pi_N) = (g_1^\# U, g_2^\# U, \cdots, g_N^\# U)$ *is a stationary Nash point (SNP) for an $N$-player continuous game if and only if:*

$$\mathbb{E}_{\omega_j \sim U, \ \forall j \neq i}[\nabla_i f_i(g_1(\omega_1), g_2(\omega_2), \cdots, g_N(\omega_N))] \equiv 0, \quad \forall \omega_i \in \mathbb{R}^d$$

*holds for all $i \in [N]$.*

From Equation (7), we also know that:

$$\delta_{g_i}(F_i)[\sigma(\omega_i)] = \langle G(\omega_i), \sigma(\omega_i) \rangle$$

In other words, the steepest direction is:

$$\delta_{g_i}(F_i) = G(\omega_i) = \mathbb{E}_{\omega_j \sim U, \ \forall j \neq i}[\nabla_i f_i(g_1(\omega_1), g_2(\omega_2), \cdots, g_N(\omega_N))]$$

Then we show the relationship between stationary Nash point and Nash equilibrium.

**Theorem 2.** *Denote $\mathcal{S}^{SNP}, \mathcal{S}^{NE}$ as the set of SNPs and NEs of a particular $N$-player continuous game. Obviously, $\mathcal{S}^{NE} \subseteq \mathcal{S}^{SNP}$. If all utility functions $f_i$ are convex, we have: $\mathcal{S}^{NE} = \mathcal{S}^{SNP}$*

*Proof.* Suppose $\pi = (\pi_1, \pi_2, \cdots, \pi_N) = (g_1^{\#}U, g_2^{\#}U, \cdots, g_N^{\#}U)$ is an SNP, we will prove it an NE when all functions $f_i$ are convex. According to the convexity and the condition of SNPs, we know that for $\forall i \in [N]$ and any other pushforward function $\tilde{g}_i$:

$$
\begin{aligned}
& F_i(g_1, g_2, \cdots, g_N) - F_i(g_1, \cdots, \tilde{g}_i, \cdots, g_N) \\
&= \underset{\omega_j \sim U, \, \forall j}{\mathbb{E}} [f_i(g_1(\omega_1), \cdots, \tilde{g}_i(\omega_i), \cdots, g_N(\omega_N)) - f_i(g_1(\omega_1), g_2(\omega_2), \cdots, g_N(\omega_N))] \\
&\geqslant \underset{\omega_j \sim U, \, \forall j}{\mathbb{E}} \left[ (\tilde{g}_i(\omega_i) - g_i(\omega_i))^T \cdot \nabla_i f_i(g_1(\omega_1), g_2(\omega_2), \cdots, g_N(\omega_N)) \right] \\
&= \underset{\omega_i \sim U}{\mathbb{E}} [(\tilde{g}_i(\omega_i) - g_i(\omega_i))^T \cdot \underset{\omega_j \sim U, \, \forall j \neq i}{\mathbb{E}} [\nabla_i f_i(g_1(\omega_1), g_2(\omega_2), \cdots, g_N(\omega_N))]] \\
&= \underset{\omega_i \sim U}{\mathbb{E}} [(\tilde{g}_i(\omega_i) - g_i(\omega_i))^T \cdot \delta_{g_i}(F_i)] = 0
\end{aligned}
\tag{8}
$$

which leads to our conclusion, that $\pi = (g_1^{\#}U, g_2^{\#}U, \cdots, g_N^{\#}U)$ is a global Nash equilibrium. $\square$

Next, we show the relationship between the zeros of Mixed-GNI function $V(g_1, g_2, \cdots, g_N)$ and SNPs of the $N$-player continuous game.

**Lemma 1.** *Assume $f : \mathbb{R}^d \to \mathbb{R}$ is a twice differentiable function, and its 1-st order gradient $\nabla f$ is $L_f$-Lipschitz continuous. Then for $\forall x, y \in \mathbb{R}^d$, we have:*

$$|f(y) - f(x) - \langle \nabla f(x), y - x \rangle| \leqslant \frac{1}{2} L_f \|y - x\|_2^2$$

*Proof.* According to the condition of $f$, there holds the following equations.

$$
\begin{aligned}
|f(y) - f(x) - \langle \nabla f(x), y - x \rangle| &= \left| \int_0^1 \langle \nabla f(x + \tau(y - x)) - \nabla f(x), y - x \rangle d\tau \right| \\
&\leqslant \int_0^1 |\langle \nabla f(x + \tau(y - x)) - \nabla f(x), y - x \rangle| \, d\tau \\
&\leqslant \int_0^1 \|\nabla f(x + \tau(y - x)) - \nabla f(x)\| \cdot \|y - x\| d\tau \\
&\leqslant \int_0^1 L_f \tau \|y - x\|_2^2 d\tau = \frac{1}{2} L_f \|y - x\|_2^2
\end{aligned}
\tag{9}
$$

$\square$

With this lemma, we can show that each global minimum of $V(g_1, g_2, \cdots, g_N)$ is also an SNP.

**Theorem 3.** *If each utility function $f_i$ is twice differentiable and its 1-st order gradient $\nabla f_i$ is $L_f$-Lipschitz continuous. Then:*

$$\frac{\lambda}{2} \|\delta_{g_i} F_i(g_1, g_2, \cdots, g_N)\|^2 \leqslant V_i(g_1, g_2, \cdots, g_N; \lambda) \leqslant \frac{3\lambda}{2} \|\delta_{g_i} F_i(g_1, g_2, \cdots, g_N)\|^2$$

*holds when $0 < \lambda \leqslant \frac{1}{L_f}$. Here, $\| \cdot \|^2$ is a functional norm which means:*

$$\|f\|^2 = \int_{[0,1]^d} \|f(\omega_i)\|_2^2 \, d\omega_i = \underset{\omega_i \sim U}{\mathbb{E}} \|f(\omega_i)\|_2^2$$

*Proof.*
$$
\begin{aligned}
& V_i(g_1, g_2, \cdots, g_N; \lambda) \\
&= F_i(g_1, g_2, \cdots, g_N) - F_i(g_1, \cdots, g_i - \lambda \delta_{g_i} F_i, \cdots, g_N) \\
&= \underset{\omega_j \sim U, \, \forall j}{\mathbb{E}} [f_i(g_1(\omega_1), g_2(\omega_2), \cdots, g_N(\omega_N)) - f_i(g_1(\omega_1), \cdots, g_i(\omega_i) - \lambda \delta_{g_i} F_i(\omega_i), \cdots, g_N(\omega_N))]
\end{aligned}
\tag{10}
$$

Then, according to Lemma 1:
$$
\begin{aligned}
& V_i(g_1, g_2, \cdots, g_N; \lambda) \\
&\leqslant \underset{\omega_j \sim U, \, \forall j}{\mathbb{E}} \left[ \lambda (\delta_{g_i} F_i(\omega_i))^T \nabla_i f_i(g_1(\omega_1), g_2(\omega_2), \cdots, g_N(\omega_N)) + \frac{L_f}{2} \lambda^2 \|\delta_{g_i} F_i(\omega_i)\|^2 \right] \\
&= \lambda \underset{\omega_i \sim U}{\mathbb{E}} \|\delta_{g_i} F_i(g_1, g_2, \cdots, g_N)(\omega_i)\|_2^2 + \frac{L_f}{2} \lambda^2 \underset{\omega_i \sim U}{\mathbb{E}} \|\delta_{g_i} F_i(g_1, g_2, \cdots, g_N)(\omega_i)\|_2^2 \\
&\leqslant \frac{3\lambda}{2} \|\delta_{g_i} F_i(g_1, g_2, \cdots, g_N)\|^2
\end{aligned}
\tag{11}
$$

And the other side of this inequality is similar. □

The theorem above tells us that, $V(g_1, g_2, \cdots, g_N; \lambda)$ is always non-negative as long as $\lambda \leqslant \frac{1}{L_f}$. And its global minima, or in the other words, its zeros, are surely SNPs, because for $\forall i \in [N]$:

$$V_i(g_1, g_2, \cdots, g_N; \lambda) = 0 \Leftrightarrow \delta_{g_i} F_i(g_1, g_2, \cdots, g_N) = 0$$

Finally, we analyze the stability of SNPs. In the following theorem, we show that the 2-nd order variation of functional $V$ is a positive semidefinite operator, which confirms the stability of SNPs.

**Theorem 4.** *The 2-nd order variation $\delta^2 V(\mathbf{g}^*; \lambda)$ is a positive semidefinite operator for $\forall \mathbf{g}^* \in \mathcal{S}^{SNP}$ and $0 \leqslant \lambda \leqslant \frac{1}{L_f}$.*

*Proof.* The 1-st and 2-nd order variation of $V_i(\mathbf{g}; \lambda)$ satisfy:

$$\delta V_i(\mathbf{g}; \lambda) = \delta F_i(\mathbf{g}) - \delta F_i(\tilde{\mathbf{g}}) + \lambda \, \delta^2 F_i(\mathbf{g}) D_i \delta F_i(\tilde{\mathbf{g}}), \tag{12}$$

where $\mathbf{g} = (g_1, g_2, \cdots, g_N), \tilde{\mathbf{g}} = (g_1, \cdots, g_{i-1}, g_i - \lambda \delta_{g_i} F_i, \cdots, g_N)$ and

$$D_i = Diag(0_{n_1 \times n_1}, \cdots, 0_{n_{i-1} \times n_{i-1}}, I_{n_i \times n_i}, 0_{n_{i+1} \times n_{i+1}}, \cdots, 0_{n_N \times n_N})$$

is a $n \times n$ matrix. Given $\mathbf{g}^* \in \mathcal{S}^{SNP}$, then $\delta F_i(\mathbf{g}^*) = 0$.

$$\begin{aligned}
\delta^2 V_i(\mathbf{g}^*; \lambda) &= \lambda \, \delta^2 F_i(\mathbf{g}^*) [2D_i - \lambda D_i \delta^2 F_i(\mathbf{g}^*) D_i] \delta^2 F_i(\mathbf{g}^*) \\
&\succeq \lambda \, \delta^2 F_i(\mathbf{g}^*) [2D_i - \lambda L_f D_i^2] \delta^2 F_i(\mathbf{g}^*) \\
&\succeq \lambda \, \delta^2 F_i(\mathbf{g}^*) D_i \delta^2 F_i(\mathbf{g}^*) \\
&= \lambda \, (\delta^2 F_i(\mathbf{g}^*) D_i)^T (\delta^2 F_i(\mathbf{g}^*) D_i)
\end{aligned} \tag{13}$$

which is positive semidefinite. Therefore:

$$\delta^2 V(\mathbf{g}^*; \lambda) = \sum_{i=1}^{N} \delta^2 V_i(\mathbf{g}^*; \lambda)$$

is also positive semidefinite. □

## 4.2 CONVERGENCE ANALYSIS

In this section, we analyze the convergence analysis of gradient descent:

$$\mathbf{g}^{(k+1)} = \mathbf{g}^{(k)} - \rho \cdot \delta V(\mathbf{g}^{(k)}; \lambda)$$

According to the definition of functional $V(\mathbf{g}; \lambda)$, it can be rewritten as the following form:

$$V(\mathbf{g}; \lambda) = \mathop{\mathbb{E}}_{\omega_j \sim U, \, \forall j \in [N]} [G_V(g_1(\omega_1), g_2(\omega_2), \cdots, g_N(\omega_N))]$$

where $G_V = \sum_{i=1}^{N} f_i(y_1, y_2, \cdots, y_N) - f_i(y_1, \cdots, y_{i-1}, y_i - \lambda \nabla_i f_i(y_1, y_2, \cdots, y_N), \cdots, y_N)$.

**Theorem 5.** *Suppose $\nabla G_V(\mathbf{x})$ is $L_G$-Lipschitz continuous. Through gradient descent, the function sequence $\mathbf{g}^{(k)}$ converges sublinearly to a stationary Nash point (SNP) $\mathbf{g}^*$ if $\rho < \frac{1}{L_G}, \lambda \leqslant \frac{1}{L_f}$.*

*Proof.* According to Lemma 1, we have:

$$\begin{aligned}
V(\mathbf{g}^{(k+1)}; \lambda) &\leqslant V(\mathbf{g}^{(k)}; \lambda) - \mathop{\mathbb{E}}_{\omega_j \sim U, \, \forall j \in [N]} \left[ \rho \, \nabla G_V((g_1(\omega_1), g_2(\omega_2), \cdots, g_N(\omega_N)) \cdot \delta V(\mathbf{g}^{(k)}; \lambda) \right] \\
&\quad + \mathop{\mathbb{E}}_{\omega_j \sim U, \, \forall j \in [N]} \frac{L_G}{2} \rho^2 \|\delta V(\mathbf{g}^{(k)}; \lambda)\|^2 \\
&= V(\mathbf{g}^{(k)}; \lambda) - (\rho - \frac{L_G}{2} \rho^2) \|\delta V(\mathbf{g}^{(k)}; \lambda)\|^2 \\
&= V(\mathbf{g}^{(k)}; \lambda) - \left( \frac{2\rho L_G - (\rho L_G)^2}{2L_G} \right) \|\delta V(\mathbf{g}^{(k)}; \lambda)\|^2
\end{aligned} \tag{14}$$

Let $k = 0, 1, \cdots, K$, and add them up, we have:

$$V(\mathbf{g}^{(K+1)}; \lambda) \leqslant V(\mathbf{g}^{(0)}; \lambda) - \left( \frac{2\rho L_G - (\rho L_G)^2}{2L_G} \right) \sum_{k=0}^{K} \|\delta V(\mathbf{g}^{(k)}; \lambda)\|^2$$

Since $\lambda \leqslant \frac{1}{L_f}$, we know that $V(\mathbf{g}^{(K+1)}; \lambda) \geqslant 0$ by Theorem 3, we have

$$\sum_{k=0}^{K} \|\delta V(\mathbf{g}^{(k)}; \lambda)\|^2 \leqslant \left( \frac{2L_G}{2\rho L_G - (\rho L_G)^2} \right) V(\mathbf{g}^{(0)}; \lambda)$$

$$\Rightarrow \min_{k \in [K]} \|\delta V(\mathbf{g}^{(k)}; \lambda)\|^2 \leqslant \left( \frac{2L_G}{2\rho L_G - (\rho L_G)^2} \right) \frac{V(\mathbf{g}^{(0)}; \lambda)}{K+1} \tag{15}$$

which completes our proof. □

## 5 EXPERIMENTS

To evaluate the practical performance of our approach, we apply it to three types of games, two-player quadratic games, general blotto games, and GAMUT games, the most popular games for evaluation of Nash equilibrium algorithms. In all the experiments, we set the local radius $\lambda = 1e\text{-}3$ and we use gradient descent as our optimization method with step size $\rho = 1e\text{-}2$ and momentum $\kappa = 0.9$. The network architecture we use for the pushforward functions $g_\theta$ is a 6-layer fully connected neural network with the size of each layer as: 20, 40, 160, 160, 40, 20. The size of its output layer is the dimension of each player's action space. From forward to backward, the activation function we use is: $\tanh, \tanh, \tanh, \text{ReLU}, \tanh, \tanh$.

We mainly compare our approach with three recent studies, gradient descent for GNI function (Raghunathan et al., 2019) (gradGNI in short), Symplectic Gradient Adjustment algorithm (Balduzzi et al., 2018) (SGA in short), and Stable Opponent Shaping (Letcher et al., 2018) (SOS in short) as they outperformed other existing algorithms applicable to continuous game settings. For all these methods, we either follow the standard hyper-parameters mentioned in the original papers, or the ones resulting in the best convergence.

### 5.1 TWO-PLAYER QUADRATIC GAME

The two-player quadratic game is defined by the the players' payoff functions $f_i$ ($i = 1, 2$):

$$f_i(\mathbf{x}) = \mathbf{x}^T Q_i \mathbf{x} + r_i^T \mathbf{x}, \tag{16}$$

where $Q_i \in \mathbb{R}^{(n_1+n_2) \times (n_1+n_2)}$, $r_i \in \mathbb{R}^{n_1+n_2}$, $\mathbf{x} = (x_1, x_2)$ and $x_i \in \mathbb{R}^{n_i}$. In our experiments, we choose $n_1 = n_2 \in \{3, 5, 10\}$. For each pair of $n_i$, we randomly generate 100 instances for the matrix $Q_i$ and $r_i$ for $i = 1, 2$. Each item in each matrix $Q_i$ and each vector $r_i$ follows the uniform distribution on $[0, 1]$ independently.

We show the converging process of all algorithms for one game instance ($n_1 = n_2 = 3$) in Fig. 1(a) as an example. As we can see, our approach effectively converges to a stationary Nash equilibrium point. While gradGNI also converge in this instance, its result has larger local regret. In other words, it obtain worse approximation to Nash equilibrium, which coincides with the essential difference between pure strategy and mixed strategy. The Mixed-GNI approach searches for the equilibrium in the mixed strategy space, which includes the pure strategy space that gradGNI searches in. On the other hand, SGA and SOS diverge in this game instance. We further take the average of the final local regrets after 2000 iterations for all the 100 instances, summarized in Tab. 1. All the algorithms show consistency as the dimension of action space increases, and Mixed-GNI outperforms others regardless of the randomness of game structures.

### 5.2 GENERAL BLOTTO GAME

We next consider the general blotto game, which differs from previous games in the action space of each player for which further constraints apply.

In a blotto game, player 1 and 2 (sometimes known as two colonels) have a budget of resource $X_1$, $X_2$ respectively. W.l.o.g we set $X_1 \leq X_2$. There are $m$ battlefields in total. In each battlefield $j$, when two players allocate $x_{1j}, x_{2j}$ resource on it, the payoff of player $i$ is:

$$U_{ij} = f(x_{ij} - x_{-ij}), \text{ where } f(\chi) = \tanh(\chi), \tag{17}$$

where $-i$ denotes the player other than player $i$. Each player's payoff across all $m$ battlefields is the sum of the payoffs across the individual battlefields. For each player $i$, a feasible pure strategy $x_i = (x_{i1}, \ldots, x_{im}) \in \mathbb{R}_+^m$ must also satisfies $\sum_{j=1}^m x_{ij} \leq X_i$. Here we adopt the generalized

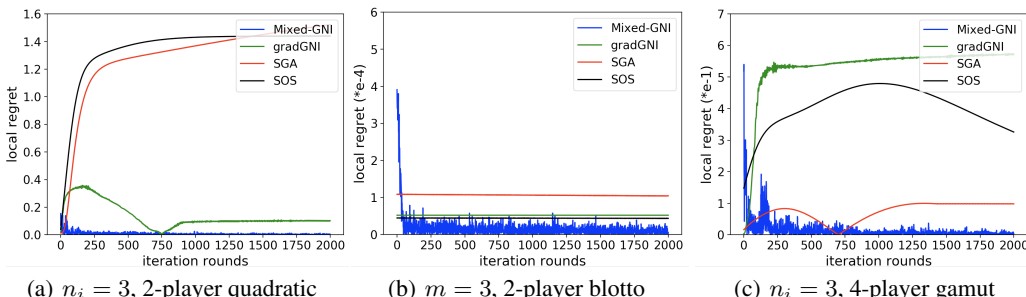

(a) $n_i = 3$, 2-player quadratic   (b) $m = 3$, 2-player blotto   (c) $n_i = 3$, 4-player gamut

Figure 1: Local Regret of Various Games.

| | Mixed-GNI (ours) | gradGNI | SGA | SOS |
|---|---|---|---|---|
| Quadratic ($n_i = 3$) | $(\mathbf{1.63 \pm 1.20})$e-3 | $(1.01 \pm 0.03)$e-1 | $2.59 \pm 0.17$ | $1.86 \pm 0.36$ |
| Quadratic ($n_i = 5$) | $(\mathbf{2.84 \pm 1.95})$e-3 | $(2.95 \pm 0.19)$e-1 | $3.92 \pm 0.22$ | $2.89 \pm 0.37$ |
| Quadratic ($n_i = 10$) | $(\mathbf{3.76 \pm 3.02})$e-3 | $(1.47 \pm 0.08)$e-1 | $2.54 \pm 0.09$ | $2.46 \pm 0.12$ |
| Blotto ($m = 3$) | $(\mathbf{6.32 \pm 4.97})$e-6 | $(2.62 \pm 0.38)$e-5 | $(5.26 \pm 0.91)$e-5 | $(7.31 \pm 0.82)$e-5 |
| Blotto ($m = 5$) | $(\mathbf{4.52 \pm 3.09})$e-6 | $(1.10 \pm 0.06)$e-5 | $(1.21 \pm 0.18)$e-5 | $(7.39 \pm 2.33)$e-6 |
| Blotto ($m = 10$) | $(\mathbf{3.62 \pm 2.39})$e-6 | $(7.60 \pm 0.49)$e-6 | $(5.94 \pm 0.26)$e-6 | $(5.02 \pm 0.28)$e-6 |
| GAMUT ($n_i = 3$) | $(\mathbf{4.95 \pm 0.42})$e-3 | $(4.80 \pm 0.81)$e-1 | $(0.94 \pm 0.13)$e-1 | $(0.96 \pm 0.17)$e-1 |
| GAMUT ($n_i = 5$) | $(\mathbf{8.90 \pm 0.79})$e-3 | $(1.52 \pm 0.27)$e-1 | $(2.59 \pm 0.60)$e-1 | $(2.67 \pm 0.81)$e-1 |
| GAMUT ($n_i = 10$) | $(\mathbf{1.54 \pm 0.86})$e-2 | $(1.84 \pm 0.48)$e-1 | $(1.76 \pm 0.32)$e-1 | $(1.90 \pm 0.47)$e-1 |

Table 1: Comparison results.

blotto game proposed by Golman & Page (2009) with continuous payoff functions. The payoff functions in vanilla blotto game is discontinuous, for which our method as well as baselines fails. In our experiments, we set $m \in \{3, 5, 10\}$. For each $m$, we randomly generate 100 instance for the budget $X_i$, following the uniform distribution on $[0, 1]$ independently.

We show the converging process of all algorithms for one game instance ($m = 3$) in Fig. 1(b) as an example. All the algorithms converges for this game, while gradGNI, SGA and SOS converge faster and more smoothly comparing with our Mixed-GNI. However, similar to the quadratic game, their final results have larger local regrets. This coincides with the fact that the mixed strategy is a better solution concept than the pure strategy, especially in blotto games. We further take the average of the final local regrets after 2000 iterations for all the 100 instances, summarized in Tab. 1. All the algorithms show consistency as the dimension of action space increases, and Mixed-GNI outperforms others regardless of the randomness of game structures.

## 5.3 GAMUT GAMES

Finally, we apply our method on the game instance generated by the comprehensive GAMUT suite of game generators designated for testing game-theoretic algorithms (Nudelman et al., 2004). GAMUT includes a group of random distributions, based on each of which the payoff of each player for each pure strategy profile can be drawn independently. To be precise, we extend the quadratic game to a multi-player version, where $r_i = 0$, and 100 game instances with 4 players are generated. For each instance, one of the distributions from the GAMUT set is selected, and each item in each matrix $Q_i$ is sampled according to it independently.

We show the converging process of all algorithms for one game instance in Fig. 1(c). Both Mixed-GNI and SGA converge, but SGA has a much worse final result than our Mixed-GNI. And this time, gradGNI diverges while SOS converges much slower. Furthermore, we take the average of the final local regrets after 2000 iterations for all the 100 instances, shown in Table 1.

From these different games, we know that our Mixed-GNI converges and performs better than two baselines in all of the three games, which shows the effectiveness and efficiency of our Mixed-GNI model. As the first algorithm to compute the mixed strategy Nash equilibrium of games with continuous action space, we believe that the technique we introduced here will enable new optimization researches of many exciting interaction domains of algorithmic game theory and deep learning.

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

## A    EXPERIMENTS ON DELTA-DIRAC GANS

Gidel et al. (2018) introduce a one-dimensional GAN where the real data follows a Dirac-Delta distribution. This experiment is also conducted in our baseline paper by Raghunathan et al. (2019). As a two-player game, the payoff functions for the two players are listed as follows:

$$f_1(x_1, x_2) = \log(1 + \theta x_1) + \log(1 + \exp(x_1 x_2))$$
$$f_2(x_1, x_2) = -\log(1 + \exp(x_1 x_2)) \tag{18}$$

where the $\theta$ is the location of the Dirac spike. In Figure 2, we show the converging process of all algorithms for one game instance (i.e. for a single $\theta$). As we can see, our Mixed-GNI and original gradGNI algorithms converge while the other two diverge. Compared with gradGNI, our approach is converging much faster. We further take the average of the final local regrets after 2000 iterations for all the 100 instances (i.e. 100 different $\theta$s) and show the results in Table 2. In this experiment, our Mixed-GNI algorithm again outperforms the others, regardless of the randomness of game structures.

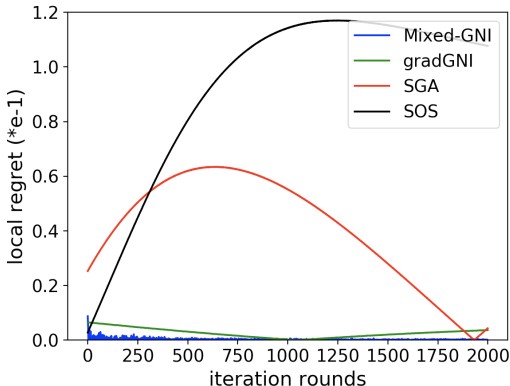

Figure 2: Local Regret of Dirac-Delta GAN Game.

|                | Mixed-GNI (ours)        | gradGNI              | SGA                  | SOS                  |
| -------------- | ----------------------- | -------------------- | -------------------- | -------------------- |
| Dirac-Delta GAN | $(\mathbf{7.23 \pm 3.70})$e-5 | $(6.52 \pm 3.98)$e-3 | $(1.39 \pm 0.96)$e-2 | $(1.41 \pm 0.87)$e-2 |

Table 2: Comparison Results

