# OpenReview forum: "Finding Mixed Strategy Nash Equilibrium for Continuous Games through Deep Learning"
_ICLR.cc/2020/Conference — Reject_

### Official Review · AnonReviewer3 · 2019-10-19
**Official Blind Review #3**

**Rating:** 6

**Review:**

This paper proposes a novel algorithm for finding mixed-strategy Nash equilibria in games with continuous action spaces. The paper proves convergence to a stationary Nash equilibrium under the assumption of convex utility functions. In quadratic games, general blotto games and GAMUT games, the proposed algorithm (MC-GNI) significantly outperforms its competitors. This experimental finding is not surprising, as its competitors were designed to compute pure-strategy Nash-equilibria.

This paper investigates an important, relatively unexplored problem. Its proposed algorithm appears theoretically justified and to achieve nontrivial performance in practice. However, I feel that there are some issues with this paper.

There has been a significant amount of recent work on differentiable games. It would benefit readers for the paper to discuss how this work and those works are related. In particular, the latter are motivated by designing gradient-based algorithms for finding parameters of neural networks corresponding to local equilibria. These neural networks may themselves be playing nondeterministic strategies in some underlying game (e.g., GANs). This closely resembles the paradigm of the algorithm proposed in this paper, which finds deterministic parameters for a network that itself induces a nondeterministic strategy. This leads to a number of questions:
1. In the experiments, the paper’s language suggests that SGA is constrained to pure strategies. Why is it not parameterized by a network producing a stochastic strategy? This seems like the sensible comparison.
2. How are the convergence guarantees discussed in this paper related to the capacity of the network being used to approximate the policy? Are they lost if the network has insufficient capacity?
3. The paper mentions that SGA outperforms other existing algorithms applicable to continuous game settings. What about stable opponent shaping??

The writing in the paper would benefit from revision. While the paper generally gets the point across, much of it feels sloppy. There are also many typos, some of which are listed below.

Typos:
- Raghunathan et al. (2019) introduces
- In precise
- drown
- for $\forall$
- we prepare each distribution πj a corresponding pushforward function
- Therefore, a common algorithm of computing
- On the one hand, global infimum can
- We develop a solution significantly improves the status-quo.
- of Mixed strategy on Continuous game

Other:
- The paper alternates between his and it when referring to players

Overall, I think this work is an outline for a nice paper but would like to see
1) Clarification regarding its relationship to existing work
2) Experimental baselines in which SGA and stable opponent shaping are parameterized to produce nondeterministic strategies
3) Cleaner writing

**Experience Assessment:**

I have read many papers in this area.

**Review Assessment: Checking Correctness Of Derivations And Theory:**

I assessed the sensibility of the derivations and theory.

**Review Assessment: Checking Correctness Of Experiments:**

I assessed the sensibility of the experiments.

**Review Assessment: Thoroughness In Paper Reading:**

I read the paper thoroughly.

---

> ### Author Response · Authors · 2019-11-08
> **Official Comment to Reviewer 3**
>
> Thank you very much for your comments. Let us answer your questions and hope it will resolve your concerns.
> (1) Question: There has been a significant amount of recent work on differentiable games. It would benefit readers for the paper to discuss how this work and those works are related.
>     Answer: The settings in differentiable game literature are different from ours. In differentiable games (e.g. defined by Balduzzi et al., 2018 and Letcher et al. 2019), the strategy of each player is represented by a parameter vector on real space, and the utility (or loss) of him is twice continuous differentiable with respect to the vector. Comparing with classical terminology defining discrete and continuous game, such parameter vector can represent a mixed strategy of discrete action space, which is a multinomial distribution on finite possible choice of actions (see the prisoner dilemma case in Letcher et al. 2019.), or a pure strategy of continuous action space, where each value of the parameter corresponds to a possible action (real number). As we all know, previous studies on equilibrium computation for differentiable games, if continuous games are considered, did not provide a way to parameterize mixed strategy of continuous action space. Meanwhile, the utility functions are not required to be twice differentiable in our work.
>
> (2) Question: In the experiments, the paper’s language suggests that SGA is constrained to pure strategies. Why is it not parameterized by a network producing a stochastic strategy? This seems like the sensible comparison.
>     Answer: The algorithm SGA, as well as stable opponent shaping, needs to compute the Hessian (second order derivative matrix) of utility functions (losses) with respect to players' strategies. In the original papers, they use each parameter to represent a pure strategy of each player for continuous game experiments, and the algorithms are used to find the (locally) optimal pure strategy. Although we are able to parameterized the mixed strategies of continuous action spaces by neural networks, the corresponding Hessian would be a huge matrix, vary hard to compute or store.
>
> (4) Question: How are the convergence guarantees discussed in this paper related to the capacity of the network being used to approximate the policy? Are they lost if the network has insufficient capacity?
>     Answer: As we all known, one should increase the size of neural networks to obtain good approximation for functions with higher dimensions. And in our experiments, the networks are not of huge sizes and they performed well. As a result, guaranteeing convergence through small networks is not of our interest in this work.
>
> (5) Question: The paper mentions that SGA outperforms other existing algorithms applicable to continuous game settings. What about stable opponent shaping (SOS)??
>     Answer: We add the comparison results with SOS in our revision. (We shall upload it soon. It shall take several days to finish all the experiments since we are comparing the average performance of 100 repeated runs.) However, we must point out that it is unrealistic to use SGA and SOS to produce mixed (nondeterministic) strategies for continuous action spaces, for the computation complexity for the Hessian of large neural networks.
>
> (6) Question: The writing in the paper would benefit from revision.
>     Answer: We modify these typos in our revision.

---

> > ### Comment · AnonReviewer3 · 2019-11-10
> > **Re: Official Comment to Reviewer 3**
> >
> > Thanks for your response. I have some additional comments below.
> >
> > (2) This clarification is what I was looking for:
> >
> > "The algorithm SGA, as well as stable opponent shaping, needs to compute the Hessian (second order derivative matrix) of utility functions (losses) with respect to players' strategies. In the original papers, they use each parameter to represent a pure strategy of each player for continuous game experiments, and the algorithms are used to find the (locally) optimal pure strategy. Although we are able to parameterized the mixed strategies of continuous action spaces by neural networks, the corresponding Hessian would be a huge matrix, vary hard to compute or store."
> >
> > I think that it would improve the clarity of the paper to have a sentence like this. It seems to me that this is a crucial point as to why solution methods for differentiable games are insufficient for the problem that this paper seeks to address.
> >
> > (6) From what I can see, these typos remain in the paper.
> >
> > In summary, (1) and (2) were my biggest concerns and I feel that the authors' response addresses these concerns adequately. I have revised my score accordingly. I urge the authors to consider spending some additional time to improve the quality of the writing.

---

> > > ### Author Response · Authors · 2019-11-13
> > > **Official Comment to Reviewer 3**
> > >
> > > Thanks again for your comments. We have modified those typos and updated the revision. The clarification for SGA/SOS and the relationship to differential games have been added in Section 2.

---

### Official Review · AnonReviewer2 · 2019-10-22
**Official Blind Review #2**

**Rating:** 6

**Review:**

The paper proposes a method to learn mixed-strategies Nash equilibrium in multi-player games. To do so they describe a gradient-descent method that aims at minimizing a Kikaido-Isoda function (which is zero if an equilibrium is found). The paper offer proofs of the convergence towards a stationary Nash equilibrium in the case of convex cost functions. It also provides an application with a strategy approximation made with a deep neural network. They authors exemplify the strengths of the method on toy problems that are quite standard in the domain.

I liked the paper very much but I have some concerns. First, I feel that the framework is based on a variational approach which would be well suited for a 0-order optimisation (like a black box or an evolutionary method). I wonder why the authors wanted to use a gradient based approach that adds a second layer of approximation and additional meta parameters to tune.

Second, I felt the theoretical proofs are not using much more than standard algebra and the convex assumption was a bit unrealistic in most of multi-agent problems. I'd like the authors to comment on this.

I was also wondering how this work can be related to other papers that try to learn a Nash equilibrium (or an \epsilon-Nash) on the bases of a difference in some norm between the value of the current policy and the value of the NE. For instance "Learning Nash Equilibrium for General-Sum Markov Games from Batch Data" by Perolat et al. This work addresses discrete action spaces but seems similar in spirit to me. Could the authors comment on this ?



**Experience Assessment:**

I have published one or two papers in this area.

**Review Assessment: Checking Correctness Of Derivations And Theory:**

I assessed the sensibility of the derivations and theory.

**Review Assessment: Checking Correctness Of Experiments:**

I carefully checked the experiments.

**Review Assessment: Thoroughness In Paper Reading:**

I read the paper at least twice and used my best judgement in assessing the paper.

---

> ### Author Response · Authors · 2019-11-08
> **Official Comment to Reviewer 2**
>
> Thank you very much for your comments. Let us answer your questions and hope it will resolve your concerns.
> (1) Question: Why we used a gradient based approach that adds a second layer of approximation and additional meta parameters to tune.
>     Answer: In our setting, we use loss function and optimization method, in order to get a mixed strategy Nash equilibrium, which is expressed as a function other than a vector. (For comparison, in discrete games, a mixed strategy is just a probability distribution on finite action choices. However in continuous games that we focus on, each player's action space is not only infinite but also continuous. Therefore, the corresponding mixed strategy is a density function, or pushforward function.) As we all know, a function space is much larger than finite-dimensional vector space. Zeroth-order optimization algorithms like evolutionary method are not able to sample functions efficiently, and it is difficult for them to quickly optimize and run over all possible functions. In our paper, we consider the neural network function space instead of the whole function space, which means we use the different neural network parameters to “quantify” all possible functions. It’s known that when our network is deep enough and wide enough, or has sufficiently many parameters, it can express almost all Lipschitz continuous functions.
>
> (2) Question: Are the theoretical proofs in this paper not using much more than standard algebra?
>     Answer: The theoretical proofs in this paper use no more than simple calculus, algebra and inequalities. However, as far as we understand, for theoretical proof in game theory or even machine learning, its viewpoint is much more important than its difficulty or mathematical tool. Specifically, the combination of “multi-player + continuous game + mixed strategy” naturally pushes variational method onto the stage, which is the first time in game theory. In our setting, the Nash equilibrium condition is much more complicated than the pure strategy one, which has been studied for a long time, but also has a very clean mathematical form. Besides, the convergence of gradient descent algorithm is also proved in the variational environment, using functional norm to bound the loss of each step, which is totally different from past related theory. We believe that the theory we proposed will have deep influence on future works of this field.
>
> (3) Question: Is the convex assumption a bit unrealistic in most of multi-agent problems?
>     Answer: The convex assumption is only required in the proof of convergence, while the algorithm itself can be applied to non-convex problems. We must point it out that in our third experiment, the multi-player games generated in GAMUT are not necessarily with convex utility functions, because our quadratic term matrix is not necessarily positive definite. Although the convergence for non-convex cases is not theoretically proved, our algorithm do have good performance on a large class of non-convex games. Since the convex assumption is very common in deep learning studies, this should not an important issue.
>
> (4) Question: How this work is related to other papers that try to learn a Nash equilibrium (or approximated) based on the difference in some norm between current policy and NE. For instance Perolat et al. ?
>     Answer: Take the paper mentioned (Perolat et al.) as an example. There are two main differences between the setting of ours and theirs. Firstly, they are considering Markov games, which contains a dynamic process, and the corresponding (weak) local Nash equilibrium is defined with respect to an expectation over possible states. On the contrast, we focus on one-shot games. Secondly, they studied games with discrete action spaces while we consider continuous ones. We want to point out that these two are quite different, as the methodology for representing strategies on discrete action spaces as well as the corresponding Bellman equation techniques is hard to be transformed for continuous spaces.

---

> > ### Comment · AnonReviewer2 · 2019-11-14
> > **Answer to authors**
> >
> > I'm not sure I understood well your answer to my comment about 0-th order methods. You say these methods cannot sample in function space but you approximate functions with neural nets. Then, I think it is clear that you can use 0-th order to optimize neural net parameters (or any parameterized function approximator). Neural nets don't involve systematically gradient descent optimisation.
> >
> > Also, convexity is indeed an assumption often made in optimization but neural nets are clearly non convex and most interesting games are not convex. So I'm not sure the proofs support very well the method. I think the convexity assumption is really too strong in this case. I'm not saying that it is not useful to know whether the algorithm converges in the ideal case, but here it makes the proofs quite simple because the algorithm is somehow based on a first order approximation of the cost function. Everything thus becomes simple if the objective function is convex.

---

> > > ### Author Response · Authors · 2019-11-15
> > > **Official Comment to Review #2**
> > >
> > > Thanks very much for your comment.
> > >
> > > (1) Zeroth order optimization can indeed be used to optimize neural network parameters. However, due to the fact that neural networks usually have a huge number of parameters, zeroth order optimization becomes less efficient and lacks theoretical guarantee. Also, zeroth order optimization is seldom used in related game theory works while gradient-based one order optimization is used everywhere. That’s why we didn’t choose zeroth order optimization algorithm.
> > >
> > > (2) As you said, convexity is an often-made assumption in both fields of machine learning and game theory and it represents an ideal case. However, without convexity assumption, one may only get some conclusion about local property. While we are able to prove our algorithm can converge to a Nash equilibrium, which is something global, with convexity assumption added. Besides, it is worth mentioned that there are some theoretically-guaranteed algorithms which only require utility functions to be twice differentiable, like SGA and SOS we used as baselines in this paper, but they can only handle pure strategy and both of them need to compute Hessian matrix, which is not practical in our case because of the huge parameter number of neural networks.
> > >     On the other hand, our proofs themselves are not simple even with the convexity assumption. We believe the way we prove the results (by using variational methods) is original in the field of Nash equilibrium computation, let alone the new setting of mixed strategy + continuous action space + multiple player. That’s our contribution.

---

### Official Review · AnonReviewer1 · 2019-10-25
**Official Blind Review #1**

**Rating:** 3

**Review:**

This work describes a method for finding mixed-strategy Nash equilibria in (normal form) games with continuous action spaces. Their work builds off (Raghunathan '19), which is a gradient-based method for finding pure-strategy Nash, based on the NI function. (It comes with all the standard caveats of gradient-based methods for finding Nash). The central contribution of this work is to parameterize a mixed strategy via a learned NN mapping from a simple distribution U[0,1]^d to the mixed strategy of interest.

My first impression on reading this work is that this learned parameterization of the mixed strategy is exactly how the GAN generator produces a distribution of samples (images). The generator network takes a noise vector sampled from U[0,1]^d or some other distribution, and samples an "action" (i.e. an image). Indeed, the original GAN paper (Goodfellow, 14) talks about this: "Generative adversarial nets are trained by simultaneously updating the discriminative distribution so that it discriminates between samples from the *data generating distribution*" (Fig. 1). It is not the case, as stated in Section 2, that GANs only produce pure strategies.

The optimization rule for GNI is different than standard update rules: specifically, there are two step sizes \lambda and \rho, if \lambda -> 0 I think it becomes steepest descent but for \lambda > 0 it is kind of like an optimistic descent method (because each agent takes a gradient step assuming the other agents have already done SD by \lambda). This update rule has some desirable theoretical properties, which I believe are mostly proven by Raghunathan '19. It would be very useful to state more explicitly which of these properties are not known to be true for e.g. independent steepest descent for each agent.

The experimental section is nice in that it shows some toy games where mixed strategy continuous equilibria do better than pure-strategy ones, and shows that MC-GNI can find something better than a pure-strategy equilibrium. But if the claim is that GNI has these good equilibrium-finding properties, and you're already using the GAN decoder strategy, why are there no experiments on GANs? Even Raghunathan has experiments on GANs (albeit ones with pure-strategy (delta-function) equilibria).

Nits:
- MC-GNI is a confusing name because MC usually refers to "Monte Carlo" in this context
- I'm confused at the end of section 3 how "implying (sic) gradient descent on these function parameters" is performed. Are points sampled from U[0,1]^d and used as an estimator for F_i, as in a GAN? I don't see how the non-MC-sampled F_i can be computed. It would be nice to clarify one way or another.
- I don't see \lambda mentioned on the RHS of the last equation in sec. 3.


**Experience Assessment:**

I have published in this field for several years.

**Review Assessment: Checking Correctness Of Derivations And Theory:**

I carefully checked the derivations and theory.

**Review Assessment: Checking Correctness Of Experiments:**

I assessed the sensibility of the experiments.

**Review Assessment: Thoroughness In Paper Reading:**

I read the paper thoroughly.

---

> ### Author Response · Authors · 2019-11-08
> **Official Comment to Reviewer 1**
>
> Thank you very much for your comments. Let us answer your questions and hope it will resolve your concerns.
> (1) Question: In our Section 2, we said that GANs only produce pure strategies. Is it true?
>     Answer: It is a typo. The GANs’ min-max structure is limited to 2-player zero-sum games. While our model can calculate the approximated mixed strategy NE of multi-player general-sum games. Basically, this is not what GAN-based models have considered. We revise it in our revision.
>
> (2) Question: It would be very useful to state more explicitly which of these properties are not known to be true for e.g. independent steepest descent for each agent.
>     Answer: The theoretical properties of the optimization is similar. With our model, we are able to solve a kind of problem which is un-solvable before. Therefore, the convergence property of optimization algorithm becomes less important in this paper.
>
> (3) Why are there no experiments on GANs?
>     Answer: In our paper, we selected current three classes of games to cover as much multi-player general-sum continuous games as possible, while GAN is regarded as a special 2-player zero-sum game. We add another experiment on GANs with the same setting as Raghunathan '19. (We shall upload the revision soon. It shall take several days to finish all the experiments since we are comparing the average performance of 100 repeated runs.)
>
> (4) MC-GNI is a confusing name because MC usually refers to "Monte Carlo" in this context
>     Answer: We change the algorithm name to “mixed-GNI” in our revision.
>
> (5) I'm confused at the end of section 3 how "implying (sic) gradient descent on these function parameters" is performed.
>     Answer: The points are indeed sampled from U[0,1]^d and used as an estimator for F_i, which is standard Monte Carlo Sampling. We clarify it in our revision.
>
> (6) I don't see \lambda mentioned on the RHS of the last equation in sec. 3.
>     Answer: It is a typo. We revise it in our revision.

---

> > ### Comment · AnonReviewer1 · 2019-11-10
> > **Clarifications**
> >
> > Thank you for your responses. I want to make sure I understand your responses (1) and (2).
> >
> > In (1) I think you are saying that GANs are used to find an equilibrium in *one particular (2p-0s)* game while you use the same modeling approach (i.e. a learned map from [0,1]^d to a mixed strategy) to find equilibria in arbitrary games (under all the standard assumptions around e.g. local minima, cycling, etc.) So GANs would be one instantiation of this method.
> >
> > In (2) I think you are saying that your optimization algorithm has no extra theoretical properties over independent steepest descent, so the important contribution is how you model mixed strategies (identically to GANs).
> >
> > So is it correct to say that the main contribution of this work is to show that the approach used to find mixed-strategy equilibria in GANs is applicable more generally to finding mixed-strategy equilibria in arbitrary games (with all the caveats of gradient descent methods)?

---

> > > ### Author Response · Authors · 2019-11-13
> > > **Re: Clarifications:**
> > >
> > > Thanks again for your comments.
> > >
> > > There are more contributions of this work, besides showing that the approach used to find mixed-strategy equilibria in GANs is applicable more generally to finding mixed-strategy equilibria in arbitrary games. We also generalize the GNI function to Mixed-GNI function which can handle mixed strategies of continuous action space and theoretically prove it is a proper loss function for gradient descent.
> > >
> > > We must point out that such a generalization as well as the proofs cannot be simply regarded as caveats of optimization. On one hand, the loss functions used in GAN approaches, e.g. Wasserstein distance, are restricted in two-player games, while our generalized GNI function can handle multiple-player because of our different formulations. On the other hand, all of our proofs are done through dealing with functional variations, which is much different from those commonly done in finite-dimensional real vector space, and has never been done before as far as we know.
> > >
> > > We believe this is also the reason why we are the first to establish a theoretical formulation of the extend mixed strategy continuous action space Nash equilibrium as a result of some optimization.

---

### Decision · Program_Chairs · 2019-12-19

**Decision:**

Reject

**Comment:**

The paper presents an algorithm to compute mixed-strategy Nash equilibria for continuous action space games. While the paper has some novelty, reviewers are generally unimpressed with the assumptions made, and the quality of the writing. Reviewers were also not swayed by the responses from the authors. Additionally, it could be argued that the paper is somewhat peripheral to the topic of the conference.¨

On balance, I would recommend reject for now; the paper needs more work.